# Smoking Prevalence, Attitudes and Behavior among Dental Students in Poland and Italy

**DOI:** 10.3390/ijerph17207451

**Published:** 2020-10-13

**Authors:** Ewa Rodakowska, Marta Mazur, Joanna Baginska, Teresa Sierpinska, Giuseppe La Torre, Livia Ottolenghi, Valeria D’Egidio, Fabrizio Guerra

**Affiliations:** 1Department of Clinical Dentistry-Cariology, University of Bergen, 5020 Bergen, Norway; 2Department of Restorative Dentistry, Medical University of Bialystok, 15-276 Białystok, Poland; 3Department of Oral and Maxillofacial Sciences, ‘Sapienza’ University of Rome, 00185 Roma, Italy; marta.mazur@uniroma1.it (M.M.); livia.ottolenghi@uniroma1.it (L.O.); fabrizio.guerra@uniroma1.it (F.G.); 4Department of Dentistry Propaedeutics, Medical University of Bialystok, 15-295 Białystok, Poland; jbaginska@wp.pl; 5Department of Prosthodontics, Medical University of Bialystok, 15-296 Białystok, Poland; teresa.sierpinska@umb.edu.pl; 6Department of Hygiene and Public Health, Sapienza University of Rome, 00185 Roma, Italy; giuseppe.latorre@uniroma1.it (G.L.T.); valeria.degidio@uniroma1.it (V.D.)

**Keywords:** Global Health Professions Student Survey, GHPSS, smoking, dental students

## Abstract

The aim of the study was to perform preliminary research to compare the smoking prevalence, attitudes and behavior between dentistry students in two universities in Europe using the standardized Global Health Professions Student Survey (GHPSS) questionnaire. This was cross-sectional carried out among dentistry students from the Medical University in Bialystok, Poland and Sapienza University of Rome, Italy. There were 582 participants; 282 were Italians, 202 were smokers and 42% were Italians. The response rate was 79.9% of Italian students and 79.6% of Polish students. The prevalence of smoking was significantly higher among Italian students (42% vs. 28.0%). Attitudes and behaviour of smokers and non-smokers differed statistically. Polish and Italian dental students presented statistically different behavior regarding the time to smoke the first cigarette, the willingness to stop smoking and trying to stop smoking in the last year. The multiple logistic regression analysis revealed that two independent variables, exposure to second-hand smoke (SHS) both at home and in public places (OR = 3.26 and OR = 5.9, respectively), showed a significantly higher occurrence of smoking. There is a high use of tobacco among dental students, which is particularly high in Italian dental students. Students realizes the positive perception of their own tobacco counsellor role in a dental setting. Dental students should be role models to their peers and patients.

## 1. Introduction

According to the World Health Organization (WHO), tobacco kills more than 8 million people each year; 7 million of those deaths are due to direct tobacco use and 1.2 million are the result of exposure to second-hand smoke [1]. Tobacco smoking is responsible for 90% of lung cancer deaths and 80% of deaths from chronic obstructive pulmonary disease (COPD) [1,2]. Moreover, it increases the risk of several diseases such as coronary heart disease, stroke, oral cancers (throat, tongue, soft palate, the tonsils) and the periodontal disease [2,3,4].

Despite many tobacco control actions being performed around the world, tobacco consumption still constitutes a global public health burden that brings huge losses to the economy and society [5,6]. Overall, between 2006 and 2017, there was a slight (6%) downward trend in the proportion of adult smokers in Europeans’ consumption of tobacco products [7], yet tobacco products still proceed to be consumed by more than a quarter of Europeans. Poland belongs to the group of countries with the highest rate of current smokers in the European Union (EU), observed at the level of 30%. Nevertheless, 28% of Poles and 24% of Italians are daily smokers. On average, Polish smokers smoke 15.7 cigarettes a day and Italians only two cigarettes fewer [7]. However, there is a large difference between the attempt to quit smoking in favor of Poland. As many as 48% of Polish smokers tried to quit compared with 24% of Italians. Nevertheless, 19% of Italians and 15% of Poles have never tried to quit smoking [7].

Based on the common risk factor (CRF) approach, oral problems have been included in the group of non-communication diseases. As smoking, which is one of the crucial CRFs, contributes to the main oral problems (oral cancer, periodontal disease, caries, halitosis), dentists are among other health professionals who should be at the front line of anti-smoking strategies. They should provide patients with the necessary information about tobacco use and the consequences on oral health outcomes [6,8,9,10]. The FDI (World Dental Federation) included in its guidelines the competencies for graduating dentists on how they need to identify and treat smoking habits [11]. So far, the vast majority of European and US/Canadian dental schools have included anti-smoking advice in their curriculum and expect students to provide some kind of tobacco counselling to patients [12,13]. Being the future generation of oral health providers, dental students’ knowledge, attitude and behavior towards smoking could be of major importance in tobacco control performance. It may impact the success or failure of any form of tobacco control activities in patients. Therefore, it is important to obtain information about their behavior and attitudes toward tobacco smoking.

The aim of the study was to perform preliminary research in order to compare the smoking prevalence, attitudes and behavior between dentistry students in two universities in Europe, the Medical University of Bialystok (MUB), Poland and Sapienza University of Rome, Italy, using the standardized Global Health Professions Student Survey (GHPSS) questionnaire. Both medical schools did not have any smoking cessation training in their curricula. We chose Polish and Italian students because these two countries differed in terms of many aspects of smoking; for example, the average age of starting smoking on a regular basis or the reasons for starting smoking [7]. Moreover, Italians are among the nationalities with the lowest rate of successful smoking cessation. As suggested by Smith et al. [13], tobacco usage in dental students usually reflects the general attitude towards smoking and may vary due to some country-specific factors. This provides an interesting background for an evaluation of students’ knowledge and attitudes and a corresponding comparison among different countries. There are surveys comparing dental students from different countries but not Polish and Italian [13,14]. Moreover, in the available literature there was no data regarding the attitudes of Polish dentistry students towards smoking. Filling this gap was one of the purposes of our survey.

## 2. Materials and Methods

### 2.1. Study Design and Study Setting

This study was cross-sectional and was carried out among dentistry students from two Faculties of Dentistry; the Medical University in Bialystok (MUB), Poland and Sapienza University of Rome, Italy. The research was conducted using the Global Health Professions Student Survey (GHPSS) questionnaire, which is a part of the Global Tobacco Surveillance System [9]. All dental students from both Universities were invited on a voluntary basis to take part in the survey between October and November 2018. The survey participation of all dental students from both universities was completely voluntary. All dental students who participated in the survey were at least 18 years old. The questionnaires were self-administered with close-ended type questions and anonymous responses. The objectives and methodology of the study were explained to the students before the survey started. The study was conducted in accordance with the protocol developed by WHO Europe and the US Centers for Disease Control and Prevention (CDC) [12], in line with the Declaration of Helsinki and after the acceptance of the Ethics Committee of the Sapienza University, Italy and the Medical University of Bialystok (MUB), Poland.

The number of students at MUB was 376 and 353 at Sapienza University of Rome. The estimated sample size was determined on the basis of nationwide data at 24% for MUB and 28% for Sapienza University of Rome. The significance level was determined at *p* = 0.05 and the maximum error at 5%. Finally, the sample size for MUB was 162 and 165 for Sapienza University of Rome.

### 2.2. Participants and Data Collection

The survey was administered using the Global Health Professions Student Survey (GHPSS) questionnaire [14,15] carried out by the WHO and the CDC. The GHPSS is a validated, anonymous, self-administrated questionnaire. The Italian version used in the study was previously validated by Chiarini et al. [16]. The original version of the GHPSS questions was translated into Polish and back-translated using the Delphi method [17] to check compatibility and accuracy with the original version. The Polish version was validated on 10 non-medical students.

### 2.3. Questionnaire

The GHPSS includes questions regarding the prevalence of tobacco use, exposure to environmental tobacco smoke (i.e., time spent with people who smoke in confined spaces), attitudes (i.e., opinions about non-smoking policies, laws, the role of health professionals (HPs) in smoking cessation), behavior/cessation (i.e., smoking habits, willingness to stop), curriculum/training (i.e., training in smoking cessation techniques on the university curriculum and knowledge about pharmacological and counselling possibilities to help smokers quit) and demographics. For the purpose of the study, we used only questions regarding demographics (age, gender, course year), prevalence of tobacco use, attitudes (opinions about no-smoking policies and laws and about the role of healthcare professionals in smoking cessation) and behavior/cessation towards smoking (smoking habit, willingness to stop, opinions about healthcare professionals who used to smoke).

### 2.4. Statistical Analysis

All data were entered into a database using Microsoft Excel then the dataset was imported into SPSS (25.0 version) (IBM, Armonk, NY, USA) for the statistical analyses. Two groups were formed and analyzed in this study; Italian and Polish students. Descriptive statistics were realized using absolute frequencies and percentages for qualitative measures on socio-demographic data, attitude and behavioral variables. A chi-squared test was performed to evaluate the differences between Italian and Polish students with a *p*-value of <0.05 taken as a threshold for statistical significance. The association between being a smoker and the potential risk factors associated such as knowledge, attitude and behavior toward tobacco use and cessation were examined using multivariate logistic regression based on the questionnaires. Odds ratios (OR) for the multivariate analysis and 95% confidence intervals (95% CI) were calculated. *p* values < 0.05 were considered to be statistically significant. The goodness-of-fit for the logistic regression model was assessed with Hosmer and Lemeshow’s test.

## 3. Results

### Description of Participants

The sample consisted of 582 participants comprising 282 Italian and 300 Polish students of dentistry. There were 202 smokers (34.7%) in examined group, out of which 59% were Italians. The response rate was similar for both countries as 79.9% of Italian and 79.6% of Polish dental students completed the survey. The majority of students were between 19 and 24 years old (73% of Italian and 86% of Polish students), with more females (66% Italian and 71% Polish) than males participating in the study. Regarding the year of attendance, 51% of Italian students were between the fourth and the sixth year of their study while 61% of Polish students ranged between the first and the third year of the course. The results are shown in Table 1.

The results regarding smoking habit, smoking cessation and exposure to second-hand smoke (SHS) are reported in Table 2 and show that Polish and Italian dental students and smokers and non-smokers differ statistically regarding smoking and cessation and the exposure to SHS. Overall, the prevalence of smoking was significantly higher among Italian students (42.0% vs. 28.0%). Moreover, fewer Polish students smoked in the previous 30 days and fewer Polish students smoked in school buildings or school properties in the previous year (<0.001). Additionally, Italian students were significantly more exposed to SHS in their houses in the previous week than Polish students (<0.001). The results showed that within a week, there was more Polish students who were statistically more often exposed to SHS between one to four days, whilst more Italians declared exposure between the last 5 to 7 days (<0.001). Similar results were found for exposure to SHS in public places in the previous week. The opinions of both groups showed that Italian and Polish dental students as well as smokers and non-smokers statistically differed regarding a specific smoking ban at universities and hospitals and whether the universities complied with it. Non-smoking and smoking students’ opinions differed statistically with non-smokers being less exposed to SHS in live places and smokers being more exposed in public places. There are statistically more smoking bans in universities and hospitals in Italy (0.001) but, on the other hand, significantly more Polish students complied with the smoking ban (<0.001). Smokers and non-smokers’ opinions differ statistically.

Table 3 describes beliefs, opinions and attitudes towards tobacco control and shows several statistical differences between the Italian and Polish dental students. Statistically, more Italian dental students stated that smoking should be banned at discos, bars (<0.001) and all enclosed public spaces (0.026), agreeing that health professionals should obtain specific training in cessation techniques (0.039). Moreover, they agreed that health professionals should advise their patients to quit smoking (0.036) and about smoking cessation (0.003). Non-smokers statistically more often agreed to a complete ban of the advertising of tobacco products, a smoking ban in restaurants, discos and bars and in all enclosed public places as well as health professionals obtaining specific training on cessation techniques, advising patients to stop smoking and using other tobacco products and to have a role in providing advice in smoking cessation. Polish and Italian dental students presented statistically different behaviors regarding the time to smoke the first cigarette, the willingness to stop smoking and trying to stop smoking in the last year.

The independent variables included in the logistic regression are shown in Table 4. The multiple logistic regression analysis revealed that two independent variables, exposure to SHS both at home and in the public places (OR = 3.26 and OR = 5.9, respectively), showed a significantly higher occurrence of smoking.

## 4. Discussion

Dentists play a key role in the prevention of all chronic conditions, not only oral diseases [11]. It is well known that smoking is one of the CRFs. Smoking dentists have been considered less predisposed to advise patients to quit this habit. However, if they change their habit regarding smoking [18], they can become a role model in tobacco cessation strategies for their patients. Unfortunately, the prevalence of smoking among dental professionals is still significant and varies from below 20% in Ireland, Japan, Spain, UK, Australia, Brazil, Canada and the USA through more than 20% in Hungary, Serbia, Italy and Romania to more than 50% in Greece [13,19,20,21,22,23]. Smoking behavior among dental students is still not a topic that is widely discussed [13]. There is no validated data concerning the smoking behavior of dental students from Poland. Therefore, the present study attempts to provide some knowledge in this field. The available literature regarding dental and medical students suggests considerable differences among countries, which are difficult to explain [13,24]. It is still not clear why social, cultural and other factors influence whether a medical student in a particular region smokes tobacco [25].

Our study demonstrated that the prevalence of a tobacco habit is paradoxically high in the population of dental students in both nationalities. However, that problem is significantly more challenging in Italy because it concerned 42% of Italian respondents compared with 28% of Polish students. Furthermore, the prevalence of smoking by dental students in Italy increased by 10% compared with the study of Pizzo et al. from 2010 [21]. The present study showed that dental students, despite their medical background, smoked tobacco more than the average rate of the general population. This proportion for Poland was 27.6% vs. 25.3% whilst Italian students smoked two times more than the general population (42.1% vs. 21.3%) according to a WHO Global Report [26]. This is in accordance with previous reports from other countries [9,10,12,22]. According to La Torre [27], there is an emergency of tobacco smoking among health professionals. He states that one of the reasons for this situation is partly attributed to not enough tobacco smoking issues in the medical curriculum. Students do not receive sufficient training on counselling patients to quit smoking [27] or, according to Cattaruzza et al., they received training but did not consider it a priority. Helping patients to quit smoking is prevention; students concentrate mostly on treatment [28]. On the other hand, it would be too simplistic to say that people from Mediterranean countries smoke more due to their lifestyle, as the results from a Spanish survey showed that only 18.3% of dental students smoked cigarettes [18]. We are not able to explain the link between the prevalence of smoking and the level of education because in Italy, the people with a higher education smoke the most (over 50%) [29] whilst in Poland, the same group smokes the least, which is fewer than 20% [30].

Italian dental students presented a more positive behavior regarding quitting smoking in the previous year in general, even though they reached for a cigarette more often than the Polish subjects. Our results demonstrated that students exposed to second-hand smoking themselves have presented a significantly higher risk of smoking. Subjects exposed to SHS in public places had almost a six-times higher chance of starting smoking, whilst those exposed in their homes were over three times more often likely to start smoking than the subjects without SHS exposure. Moreover, we found that 83% of Italian and 29% of Polish students were exposed to second-hand smoking in public places and 44% and 30% of Italian and Polish subjects were exposed to it at home, respectively. Therefore, SHS could be a reason for the higher percentage of Italian than Polish smokers. It is highly probable that environmental factors constitute the major determinant of smoking prevalence in the evaluated populations. Another finding supporting this assumption is that despite the differences in smoking prevalence, students in both locations were equally aware of the risks posed by smoking and equally supportive of restrictions on smoking in public places and the value of health professionals serving as role models for their patients. Thus, while their knowledge of and attitudes towards smoking were similar, their smoking-related behaviors were significantly different.

The high prevalence of tobacco habits and the high risk of SHS among dental students in both countries indicate the urgent need to reduce smoking in their environment. However, such a reduction could be difficult to reach due to a high rate of so-called social smoking in young people [31]. According to Cristakis et al. [32], smokers have many other smokers in their social network. Peer influence is the main reason for starting smoking followed by personal attitude and stress [7,33]. Moreover, the influence of peer smoking increases and becomes stronger than parental smoking [34]. Students are prone to smoke with companions; for example, during breaks between classes. Cigarettes can also be used to control stress connected with classes and exams, which is presumed to be a reason for a smoking habit [20,35]. Eating out frequently, which is typical for students, is associated with a higher probability to become a smoker [36]. Furthermore, it has been found that as many as two-thirds of young tobacco users have at least one smoking relative, making a family background a relevant factor of a smoking habit.

The level of acceptance of anti-tobacco regulations was very high in both nations. Despite the fact that a smoking habit was more widespread among Italian than Polish students, Italians were more prone to accept a ban of smoking in all public places. However, the students’ declarations, especially in Italy, contradicted their personal behavior. Almost 29% of Italian and 17% of Polish students smoked on school premises/properties during the previous year and 16% and 5.5% of them smoked in school buildings, respectively. Moreover, at the universities and the hospitals in both countries, students encountered specific smoking bans that were followed by half of the Italian students and over 80% of students in Poland. On the other hand, the Italian students’ attitude towards advice to patients to quit smoking and the role of health professionals in providing advice about smoking cessation was significantly stronger than in Polish dental students. What was expected was that non-smokers supported almost all issues regarding the prevention and treatment of a smoking habit. It cannot be clearly determined whether the smoking behavior of the health professionals or smoking advice from them influenced more smoking cessation advice to their patients. These are complex decisions that are influenced not only by smoking status but also by age, perception of the self as a role model, concerns for a patient relationship, training in cessation treatment and attitudes towards smoking [37].

According to Prakash et al. [38], dental students can play an important role in the promotion of tobacco cessation in their patients, which in contrary to other surveys, suggest that medical, nursing and pharmacy students have a significantly higher chance of being taught to support smoking cessation for patients [39]. In our study, smokers and non-smokers presented a statistically different approach to the majority of issues regarding tobacco control. Non-smokers’ support for the ban on tobacco use in public spaces was in agreement with previous observations from other countries [40,41,42]. Similar results were shown by Warren et al. [14], although other studies found that non-smokers presented a higher level of awareness regarding tobacco control [22,23,36,43].

Our study is extremely preliminary and additional research is needed to clarify the determinants of smoking among dental students in the various locations and the actions needed to reduce tobacco use in this health profession. The questionnaire did not differentiate smoking from e-smoking. Many smokers have recently switched to e-cigarettes because vaping is perceived to be less harmful to health and is considered a step to smoking cessation [44,45,46]. This aspect has to be considered in future research to better understand the prevalence data and differences between populations. We did not differentiate such tobacco products as cigars, pipes, water pipes, oral, chewing and nasal tobacco. However, according to EU data [7], their consumption in Poland and Italy is at a very low level. Furthermore, we did not separate those who have never smoked from past smokers. Such division could help better understand the patterns of smoking cessation. Another limitation is the opinion of non-respondents. Although the response rate was significant in both examined groups, there were approximately 20% of Italian and Polish students who did not participate. We do not know whether this was due to their refusal or for any other reason, or if they were smokers or non-smokers.

## 5. Conclusions

There is a high tobacco use among dental students, which is particularly high in Italian dental students. Dental students should be role models to their peers and patients as well as becoming the target group of the tobacco control efforts. They can implement tobacco cessation intervention to provide and assist their patients in limiting tobacco usage. Students realize the positive perception of their own tobacco counsellor role in a dental setting.

## Figures and Tables

**Table 1 ijerph-17-07451-t001:** Description of the sample.

Group	Italian Students N (%)	Polish Students N (%)	Total
Sample	282	300	582
Age (years)			
15–18	8 (2.8)	2 (0.6)	10
19–24	207 (73.4)	257 (85.7)	464
≥25	67 (23.8)	41 (13.7)	108
Sex			
Male	96 (34.0)	86 (28.7)	182
Female	186 (66.0)	214 (71.3)	400

**Table 2 ijerph-17-07451-t002:** Smoking and exposure to second-hand smoke (SHS) in dental students.

Investigated Area		Italy	Poland	*p*	Total	Non-Smoker	Smoker	*p*
Smoking and cessation								
Smoked in past 30 days	no	163 (58.0)	217 (72.0)	<0.001	380	/		/
	yes	119 (42.0)	83 (28.0)		202			
Smoked on school premises/property past year	no	129 (46.0)	54 (18.0)	<0.001	183			
	yes	81 (29.0)	50 (17.0)		131			
	I never smoked	72 (25.0)	196 (65.0)		268			
Smoked in school building past year	no	162 (57.0)	88 (29.0)	<0.001	250			
	yes	44 (16.0)	14 (5.0)		58			
	I never smoked	76 (27.0)	198 (66.0)		274			
Exposure to second-hand smoke (SHS)								
Exposure to SHS in the house in past week	Never	158 (56.0)	210 (70.0)	<0.001	368	283 (74.0)	85 (42.0)	<0.001
	1–2 days	40 (14.0)	41 (14.0)		81	46 (12.0)	35 (17.0)	
	3–4 days	15 (5.5)	20 (7.0)		35	18 (5.0)	17 (8.5)	
	5–6 days	18 (6.5)	7 (2.0)		25	10 (3.0)	15 (7.5)	
	7 days	51 (18.0)	22 (7.0)		73	23 (6.0)	50 (25.0)	
Exposure to SHS in public places in past week	Never	48 (17.0)	94 (31.3)	<0.001	142	125 (33.0)	17 (8.5)	<0.001
	1–2 days	51 (18.0)	100 (33.4)		151	119 (31.0)	32 (16.0)	
	3–4 days	49 (17.0)	48 (16.0)		97	57 (15.0)	40 (20.0)	
	5–6 days	41 (15.0)	16 (5.3)		57	38 (10.0)	19 (9.5)	
	7 days	93 (33.0)	42 (14.0)		135	41 (11.0)	94 (46.0)	
Are there specific smoking bans in university and hospital environments?	no	15 (5.0)	41 (14.0)	0.001	56	43 (11.0)	13 (6.0)	0.021
	yes, both in university and hospital	218 (77.0)	217 (72.0)		435	278 (73.0)	157 (78.0)	
	yes, only in hospital	27 (10.0)	14 (5.0)		41	25 (7.0)	16 (8.0)	
	yes, only in university	22 (8.0)	28 (9.0)		50	34 (9.0)	16 (8.0)	
Does your university comply with the smoking ban?	no	130 (46.0)	25 (8.3)	<0.001	155	91 (24.0)	64 (32.0)	0.002
	no, there is no ban	15 (5.0)	25 (8.3)		40	32 (8.0)	8 (4.0)	
	yes	137 (49.0)	250 (83.4)		387	257 (68.0)	130 (64.0)	

Chi-squared test.

**Table 3 ijerph-17-07451-t003:** Dental students’ beliefs, opinions and attitude toward tobacco control.

Investigated Area		Italian	Polish	*p*	Total	Non-Smoker	Smoker	*p*
Should tobacco sales to adolescents be banned?	no	22 (8.0)	24 (8.0)	0.92	46	31 (8.0)	15 (7.0)	0.023
	yes	260 (92.0)	276 (92.0)		536	349 (92.0)	187 (93.0)	
Should there be a complete ban of the advertising of tobacco products?	no	61 (22.0)	50 (17.0)	0.128	111	68 (18.0)	43 (21.0)	0.015
	yes	221 (78.0)	250 (83.0)		471	312 (82.0)	159 (79.0)	
Should smoking be banned in restaurants?	no	10 (3.5)	15 (5.0)	0.38	25	5 (1.0)	20 (10.0)	<0.001
	yes	272 (96.5)	285 (95.0)		557	375 (99.0)	182 (90.0)	
Should smoking be banned in discos/bars/pubs?	no	37 (13.0)	91 (30.0)	<0.001	128	64 (17.0)	64 (32.0)	<0.001
	yes	245 (87.0)	209 (70.0)		454	316 (83.0)	138 (68.0)	
Should smoking be banned in all enclosed public places?	no	9 (3.0)	22 (7.0)	0.026	31	14 (4.0)	17 (8.0)	0.001
	yes	273 (97.0)	278 (93.0)		551	366 (96.0)	185 (92.0)	
Should health professionals (HPs) obtain specific training on cessation techniques?	no	16 (6.0)	31 (10.0)	0.039	47	31 (8.0)	16 (8.0)	0.024
	yes	266 (94.0)	269 (90.0)		535	349 (92.0)	186 (92.0)	
Should HPs serve as role models for their patients and the public?	no	64 (23.0)	76 (25.0)	0.45	140	83 (22.0)	57 (28.0)	0.06
	yes	218 (77.0)	224 (75.0)		442	297 (78.0)	145 (72.0)	
Should HPs routinely advise patients to stop smoking?	no	15 (5.0)	16 (5.0)	0.99	31	11 (3.0)	20 (10.0)	<0.001
	yes	267 (95.0)	284 (95.0)		551	369 (97.0)	182 (90.0)	
Should HPs advise patients who use other tobacco products to quit using these products?	no	13 (5.0)	27 (9.0)	0.036	40	21 (5.5)	19 (9.0)	0.005
	yes	269 (95.0)	273 (91.0)		542	359 (94.5)	183 (91.0)	
Do HPs have role in providing advice or information about smoking cessation to patients?	no	12 (4.0)	32 (11.0)	0.003	44	25 (7.0)	19 (9.0)	0.012
	yes	270 (96.0)	268 (89.0)		538	355 (93.0)	183 (91.0)	
Are patient chances of quitting smoking increased with advice from HPs?	no	49 (17.0)	60 (20.0)	0.417	109	62 (16.0)	47 (23.0)	0.074
	yes	233 (83.0)	240 (80.0)		473	318 (84.0)	155 (77.0)	
**Behavior/cessation**								
After what time do you smoke your first cigarette?	in 30 min	6 (2.0)	9 (3.0)	<0.001	15	/		
	after 30 min	73 (26.0)	20 (7.0)		93			
	now I do not smoke	84 (30.0)	50 (17.0)		134			
	I have never smoked	119 (42.0)	221 (73.0)		340			
Would you like stop smoking?	no	22 (8.0)	14 (5.0)	<0.001	36			
	yes	59 (21.0)	19 (6.0)		78			
	I have never smoked	118 (42.0)	222 (74.0)		340			
	now I do not smoke	83 (15.0)	45 (15.0)		128			
Did you try to stop smoking in the last year?	no	48 (17.0)	21 (7.0)	<0.001	69			
	yes	59 (21.0)	33 (11.0)		92			
	I did not smoke last year	52 (18.0)	15 (5.0)		67			
	I do not smoke	123 (44.0)	231 (77.0)		354			

Chi-squared test.

**Table 4 ijerph-17-07451-t004:** The association between being a smoker and the potential risk factors using multivariate logistic regression.

Independent Variables	Categories of Independent Variables	OR (95% CI)
Age	other*/19–24	1.1 (0.6–2.2)
Gender	Female (reference)*/male	1.1 (0.62–2.02)
Year of attendance	1–3 (reference)*/4–6	0.59 (0.35–1.02)
Exposure to SHS in live places in past week	never*/from 1 to 7 days	3.26 (1.9–5.6)
Exposure to SHS in public places in past week	never*/ from 1 to 7 days	5.9 (2.1–16.1)
Are there specific signs about a smoking ban in university and hospital environments?	no*/yes, both in university and hospital	1.6 (0.7–3.7)
	no*/yes, only in hospital	1.5 (0.6–3.7)
	no*/yes, only in university	3.6 (0.6–22)
Does your university comply with the smoking ban?	no*/no, there is no ban	1.4 (1.03–1.9)
	no*/yes	0.9 (0.5–1.7)
Should tobacco sales to adolescents be banned?	no*/yes	2.08 (0.6–6.5)
Should there be a complete ban of the advertising of tobacco products?	no*/yes	0.9 (0.4–1.9)
Should smoking be banned in restaurants?	no*/yes	0.15 (0.02–1.2)
Should smoking be banned in discos/bars/pubs?	no*/yes	0.6 (0.25–1.5)
Should smoking be banned in all enclosed public places?	no*/yes	1.4 (0.26–7.4)
Is it fair that health professionals have specific training in smoking cessation techniques?	no*/yes	1.1 (0.3–4.2)
Should HPs serve as role models for their patients and the public?	no*/yes	0.9 (0.4–1.8)
Should HPs routinely advise patients to stop smoking?	no*/yes	0.3 (0.07–1.22)
Should HPs regularly advise their patients to stop taking tobacco (chewing, sniffing) or smoking cigars or pipes?	no*/yes	1.5 (0.22–10.7)
Do HPs have role in providing advice or information about smoking cessation to patients?	no*/yes	1.3 (0.23–7.11)
Are patient chances of quitting smoking increased with advice from HPs?	no*/yes	1.8 (0.8–3.8)

Odds ratios (OR) for the multivariate analysis and 95% confidence intervals (95% CI) were calculated. *p* values < 0.05 were considered to be statistically significant. Results of the logistic regression methods. Unadjusted estimates.

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
