# Peer review of "Smoking Prevalence, Attitudes and Behavior among Dental Students in Poland and Italy"

_ijerph, 2020, doi:10.3390/ijerph17207451_

Round 1

Reviewer 1 Report

Thank you for the opportunity to review the manuscript titled, “Smoking prevalence, attitudes and behaviour among dental students in Poland and Italy using the Global Health Professions Student Survey". This study compare the smoking prevalence, attitudes and behaviour between dentistry students in two universities in Europe, but there are limitations that limit the utility of the study. 

The Global Health Professions Student Survey was used as a questionnaire for the study, but I don't think it should go in the title of the article.

In the material and methods section, more information is needed about The Global Health Professions Student Survey used, how that questionnaire is measured.

In lines 79-80 it is mentioned that the study has the approval of the Ethics Committee of the Sapienza. The registration or identification number of that approval by the ethics committee would be missing.

In the references, in lines 248-252, the first two references are not well referenced.

Author Response

Thank you so much for the in-depth review of our article. Your comments helped us improve our manuscript for the publication.

1 We have changed the title according to the Reviewer’s suggestion. The current title is: Smoking prevalence, attitudes and behaviour among dental students in Poland and Italy.

2 In the Material and Methods section, we added more information about the Global Health Professions Student Survey.

The GHPSS includes questions regarding: prevalence of tobacco use; exposure to environmental tobacco smoke (i.e., time spent with people who smoke in confined spaces);  attitudes (i.e., opinions about non-smoking policies, laws, the role of HPs in smoking cessation;  behavior/cessation (i.e., smoking habits, willingness to stop);  curriculum/training (i.e., training in smoking cessation techniques on the university curriculum and knowledge of medication and counselling helping smokers to quit);  demographics.

3 The Ethical Committee of Sapienza University of Rome, number: 4991/2018.

4 The first two references were corrected according to the suggestions.

Reviewer 2 Report

This study compared the smoking practices and attitudes of Polish and Italian dental students. As the authors correctly point out, dental professionals, as with any health professional, can play an important role in encouraging their patients to quit smoking. This did seem to be the majority view of the students in this study as well (fortunately). I was confused with some aspects of this manuscript.

  1. Why was there a comparison between Polish and Italian students? There are some differences in the country smoking rates but Im not sure if that is large enough to justify a comparison between the countries. It would make sense to compare countries if they had different smoking cessation training in their curricula, but there was no information on this.

  1. Most students (Polish, Italian, smokers and non-smokers) agreed with tobacco control policies, including the role HPs have in advising patients. There were some statistical differences but even in these cases the students were overwhelming in favour in tobacco control practices. This is important and promising. The authors need, therefore, to determine if it is actually the smoking behaviour of the HP or the smoking advice from the HP that is more important for helping patients quit. They should see if any of the literature discusses this

  1. On p8 (lines 174/175) the authors comment that ‘..the disproportion between Italian students and the general population is difficult to interpret.’ However, this seems to be answered at the end of the same paragraph (lines 185/186) with the statement that ‘..in Italy it is the people with higher education that smoke the most (over 50%)..’. I assume dentistry students are generally in the ‘higher education’ group.

  1. In the conclusion the authors state that ‘…cessation training should be integrated in the dental curricula, and dental students should be trained in cessation techniques. However, they have not provided any evidence that it is NOT integrated. Admittedly, they do reference a couple of papers from 2013 (refs 24 and 25) suggesting that training may be limited or given low priority, but it would be good if the authors had explored the current medical curricula for the two countries to confirm that tobacco cessation training was still poorly taught.

Minor issues

  1. In the Introduction the authors report the smoking prevalence in Poland and Italy and then the proportion of daily smokers. What is the difference (that is what metric is used to determine smoking prevalence that is different to the proportion of daily smokers)? Also, apparently there is a 4% increase in daily smokers in Italy, but compared to what?

  1. Second sentence of section 3.1 (line 109) the authors state ‘There were 202 smokers (34.7%) in examined group, out of which, 42% were Italians.’ Actually 59% of smokers were Italian (42% of Italians smoked). This needs to be amended
  2. The authors use the word ‘proved’ a couple of times in the Discussion (lines 168 and 190). This study did not actually prove anything. This word needs to be changed to something like ‘demonstrated’, ‘found’, or ‘showed’

Author Response

Thank you so much for the in-depth review of our article. Your comments helped us improve our manuscript for the publication.

1 We added an explanation regarding the comparison of Polish and Italian dental students in the Introduction section as suggested:

Both Polish dental students from the Medical University of Bialystok, Poland, and Italian students form the Sapienza University of Rome, Italy, did not have any smoking cessation training in their curriculum. There are surveys comparing Italian dentistry students with students from other countries, but not with Poland. According to the Eurobarometer report [7], smoking is similarly common in Southern and Eastern Europe, but Italians are among nationalities with the lowest rate of successful smoking cessation.

2 We added sentences in reply to the Reviewer’s question in the text:

It cannot be clearly determined whether smoking behavior of health professionals or smoking advice from them influenced more smoking cessation advice to their patients. These are complex decisions which are influenced not only by smoking status, but also by age, perception of self as a role model, concerns for patient relationship, training of cessation treatment and attitudes towards smoking [34].

3 We removed this sentence.

4 We have decided to remove this sentence because dental students from the Medical University of Bialystok, Poland, and the Sapienza University of Rome, Italy, did not have any smoking cessation training in their curriculum  and we cannot provide any evidence regarding cessation training at these universities.

Minor issues

5   We removed an unnecessary sentence.

6 The percentages were corrected.

7 In the Discussion section, the word “proved” was changed to “demonstrated” as suggested.

Reviewer 3 Report

This is a useful and interesting paper that could be made much more useful, more interesting and of greater practical value with some modest re-framing and addition of some additional external data.

Proposed reframing:

The paper, as currently written, asserts that dentists should model healthy behavior by not smoking (or vaping). It laments the fact that this is not so and suggests adding anti-tobacco content to dental school curricula. By comparison, it shows that, while there are substantial differences in smoking prevalence in two dental schools, this sad situation and proposed solution is the same for both of them.

This reviewer recommends reframing the paper as follows: 

  1. First ask what are the determinants of smoking behavior in dental students in each of the two countries.
    1. The obvious determinant is the prevalence of smoking in each of the cities or countries, expressed separately for men and women, and, for this analysis, restricted to persons about 19-24 years of age.
    2. This reviewer urges restricting this part of the analysis to the 19-24 age group and showing the data separately by gender for each country. Comparable data should be available on-line from the local or national health department. This should confirm or deny the impression that the major determinant of dental student smoking prevalence is the city or national prevalence. This would show the degree to which dental students in each of the two countries are comparable to their age-matched peers in each city or country. (if the comparable city or national data are not readily available, I suggest contacting Professor Polosa, as noted in “3,”below.
    3. With that as a starting point, authors then need to consider the issue of e-cigarettes. Here the year each survey was done is critical. Even though the dental school survey did not differentiate vaping from smoking, this differentiation should be available in the city or national data.
    4. Vaping is not smoking and should never be referred to as smoking. Vaping involves little or no combustion, presents less than 5% the risk of potentially fatal disease than smoking, and, with the possible exception of the high-dose JUUL product, is almost certainly less addictive.
      1. Another consideration is that much of vaping (per the American experience) is vaping flavor-only, with no nicotine, or the vaping of marijuana. The failure of the dental school survey data to differentiate smoking from vaping is a major weakness of this study and needs to be acknowledged as such.
      2. Almost all vaping is by past or current smokers. While non-smokers frequently experiment with e-cigarettes, or occasionally use them in social settings, it is rare for a non-smoker to vape 20 or more days per month. Thus, e-cigarettes (except for the high-dose JUUL product) do not addict non-smokers. Instead smokers satisfy their urge to smoke with these much less harmful and easier to quit products. In other words, at risk of some over-simplification, smoking addicts and kills, vaping does not. Vaping and smoking are not the same.
  • For more background on the comparison of smoking to vaping, I urge consideration of a paper I wrote in 2014, with all findings as true today as they were then (a paper published in this same IJERPH). https://www.mdpi.com/1660-4601/11/6/6459
  1. If your data allow, I would restrict the rest of the analysis to students who have smoked in the past 30 days, and compare never-smokers to ever smokers to current smokers as evidence (or lack of evidence) of awareness of the harms caused by smoking.
  2. When discussing the limitations of the dental school data-sets, I urge referencing the prevalence of use of other combustion-related tobacco products (pipe’s cigars, hookhas) and smokeless products (chewing tobacco, snus, etc). In the USA, these products represent about 20% of tobacco use. From the perspective of this paper, the combustion-related products carry a risk similar to that of cigarettes, while snus carries risks similar to e-cigarettes. The risks posed by chewing tobacco vary considerably by product and are therefore beyond the scope of this discussion – but this issue should be acknowledged as something to be considered in future dental student surveys.
  1. If your data then show that both smokers and non-smokers, male and female, in both countries support limitations on smoking – you will know that they already recognize the health hazards, but not enough to counter their craving for nicotine. If that is the case, the proper response of the dental schools should not be adding something to the curriculum, but offering and urging use of dental-school-sponsored smoking cessation programming.
  2. If the comparable city and national data, are not available, I suggest contacting Professor Riccardo Polosa, MD, PhD, at the Center of Excellence for the acceleration of Harm Reduction (CoEHAR) at the University of Catania (Italy) Tel (+39) 095.478.1124; e-mail: [email protected]. I feel certain he will be happy to assist you.
  3. Comments on sections of the current text (not addressed above):
    1. Introduction – Do not use the terms “Tobacco use” and “smoking as synonymous.  Outside Asia and Africa, all of the tabulated tobacco-related mortality data is due to a single product- the combustible cigarette.
    2. Introduction - When comparing Italy to Poland, I urge, if and as possible, comparisons based on young adults, separately by gender, and with consideration of e-cigarette use. I think that there is a high probability that the higher prevalence of smoking in Italy in your current text probably represents a difference in consumption of e-cigarettes, not cigarettes.
    3. In your Materials and Methods section, it is critical to note the year the dental student surveys were done, as both smoking prevalence and vaping prevalence have substantially changed year to year during this past decade. It is also important to sort out, by gender, current and ever smoking, cigarettes smoked per day, number of days of the month for smoking – to the extend allowed in your dental school data set.
    4. If your data show significant differences in smoking prevalence by gender (male/female) or by age group (15-18, 19-24, 25+) you may want to consider presenting the data in your tables separately for these groups, or just restrict it to the 19-24 age group. Use of e-cigarettes is much more variable by age than cigarettes, with the younger groups vaping more.

Author Response

Thank you so much for the in-depth review of our article. We applied any re-framing that was possible in such a short time given for the review. Unfortunately, in some cases we did not collect the data you requested, therefore no changes were possible. For instance, the GHPSS questionnaire, which is a validated tool designed by WHO, does not comprise questions regarding vaping so such an interesting comparison between smoking and vaping was not possible. Besides, the aim of the study was to perform preliminary research in order to compare smoking prevalence, attitudes and behavior between dentistry students in two universities in Europe using the standardized Global Health Professions Student Survey (GHPSS) questionnaire.

1 1 Age and gender did not show any significantly higher occurrence of smoking (as shown in Table 4 presenting the data of multivariate logistic regression). That is also why we did not present the data depending on gender and age in a separate table. We did not restrict the age of students to only 19-24 years. The aim of our study was to compare dental students of two dental universities as a whole group. Even the title suggests that the study group consisted of all dental students, and not of a group of students of certain age.

2  We cannot confirm that the major determinant of dental student behavior was the prevalence of smoking in each city because in both universities students came from different locations. In our manuscript, we have already compared national smoking prevalence to smoking prevalence of dental students. Besides, the participants in our survey were not matched by age or gender.  

3, 4, 1, 2 The GHPSS is a validated questionnaire  designed by WHO and Centers for Disease Control and Prevention. It is a questionnaire about smoking. It does not include questions regarding vaping and such information was included as a limitation of the study (line 228). Furthermore, we did not collect the data regarding vaping and we are not able to compare it. The Reviewer asked us to change a carefully prepared article using data that is not available to us.

 The Reviewer’s suggestions will be used while planning our next survey. We were not able to include the data in the article because we did not collected such data. Besides,  the title is: “Smoking prevalence..….”. As correctly mentioned by the Reviewer, smoking and vaping is not the same. It was not an aim of our study to compare smoking and vaping, and it cannot be considered as a weak point.

1 The data did not allow us to compare ever-smokers to current smokers, but only current smokers to non-smokers

2 Unfortunately, we did not collect any data on pipe's cigars, hookahs, chewing tobacco, snus etc. and we are unable to discuss these issues. However, in evaluated countries, the prevalence of such tobacco products was insignificant (0-2%). We added such information in the paragraph referring to the limitation of the study.

2  Thank you very much for this comment. In fact, our data showed that both smokers and non-smokers supported limitations on smoking, however, this attitude was stronger in the non-smoker group. Both Polish dental students and Italian students did not have any smoking cessation training in their curriculum.

3 Thank you for suggesting a contact with prof Polosa.

Response to comments on sections:

1 We changed the terms where it was possible. We would like to emphasize that our terminology was mainly taken from WHO website and Centers for Disease Control and Prevention website

2 Thank you for your comments, but we did not collect data regarding e-cigarettes. It was not the aim of our study to compare smoking with vaping. The GHPSS is a validated tool that does not comprise questions regarding e-cigarettes. Thus we were not able find whether higher prevalence of smoking in Italy resulted from the consumption of e-cigarettes, not cigarettes. However, your line of thinking is very interesting and we will use it in our further research. In Table 4 with multivariate logistic regression, age and gender did not show any significantly higher occurrence of smoking. That is why we did not have separate gender and age groups in a separate table. We did not restrict the age of students to only 19-24 years. The aim of our study was to compare all dental students of these universities, not restricted to any age group

3 The data did not allow us to compare ever-smokers to current smokers, but only current smokers to non-smokers. The vaping data was not collected in our study as it was not a subject of our study. The survey took place between October and November 2018.

4 The analysis did not show any association between being a smoker and age and gender. The restriction to age to 19-24 would exclude almost 20% of all participants and would result in the failure to achieve our aim to compare smoking prevalence, attitudes and behavior between dentistry students (future oral health providers) from two dental schools located in different countries. We did not collect any data regarding e-cigarettes. That was not the aim of our study.

Round 2

Reviewer 3 Report

This paper is significantly improved, but a few more changes are in order. Some of the notes in the author response to the reviewer should be reflected in the paper.

First, two very small points of wording:  1) in Table 2, the term “live places” should be replaced with the term “the house” so the same wording appears n the table and the narrative (line 134).  2) line 245, the term “e-smoking” should be replaced with “smoking from the vaping of e-cigarettes.” 

Line 237:  The phrase “Our study displays some limitations” should be replaced with “Our study is extremely preliminary and additional research is needed to clarify the determinants of smoking among dental students in the various locations and the actions needed to reduce tobacco use in this health profession.”

The sentence beginning on line 246 “Currently, many people . . . “Should be reworded as follows:  “Many smokers have recently switched to e-cigarettes because vaping is perceived as less harmful to health and is considered a step to smoking cessation {44,46}.” Thus, differentiating smoking from vaping is needed to better understand the prevalence data and differences between the two schools.  Equally important is the differentiation of never smokers from past smokers, to better understand patterns of smoking cessation between the two schools.

In the various sections of the paper, two issues need more emphasis.

The first is the finding that the difference in smoking exposure in the home and public places correlated with the differences in smoking prevalence between the two schools. This suggests that environmental factors may be a major determinant of smoking prevalence in these student populations.

The second major finding is that, despite these differences, students in both locations were equally aware of the risks posed by smoking and equally supportive of restrictions on smoking in public places and the value of health professionals serving as role models for their patients. Thus, while their smoking-related behaviors were significantly different between the two sites, their knowledge and attitudes about smoking were not.

Additional research will be of value to resolve these seemingly conflicting findings and to improve the smoking-related behavior of both dental students and the clients they serve.

Author Response

We are very grateful to Reviewer for the careful analysis of our paper and valuable comments.

1 First, two very small points of wording:  1) in Table 2, the term “live places” should be replaced with the term “the house” so the same wording appears n the table and the narrative (line 134).  2) line 245, the term “e-smoking” should be replaced with “smoking from the vaping of e-cigarettes.” 

Suggested changes have been made

2 Line 237:  The phrase “Our study displays some limitations” should be replaced with “Our study is extremely preliminary and additional research is needed to clarify the determinants of smoking among dental students in the various locations and the actions needed to reduce tobacco use in this health profession.”

Suggested change has been made

3 The sentence beginning on line 246 “Currently, many people . . . “Should be reworded as follows:  “Many smokers have recently switched to e-cigarettes because vaping is perceived as less harmful to health and is considered a step to smoking cessation {44,46}.” Thus, differentiating smoking from vaping is needed to better understand the prevalence data and differences between the two schools.  Equally important is the differentiation of never smokers from past smokers, to better understand patterns of smoking cessation between the two schools. 

Suggested change has been made

In the various sections of the paper, two issues need more emphasis. Add to discussion

4 The first is the finding that the difference in smoking exposure in the home and public places correlated with the differences in smoking prevalence between the two schools. This suggests that environmental factors may be a major determinant of smoking prevalence in these student populations.

We added this valuable comment in the 3rd paragraph of Discussion 

5 The second major finding is that, despite these differences, students in both locations were equally aware of the risks posed by smoking and equally supportive of restrictions on smoking in public places and the value of health professionals serving as role models for their patients. Thus, while their smoking-related behaviors were significantly different between the two sites, their knowledge and attitudes about smoking were not.

We added this valuable comment in the 3rd paragraph of Discussion 

Additional research will be of value to resolve these seemingly conflicting findings and to improve the smoking-related behavior of both dental students and the clients they serve.

The statement about the need for additional research has been emphasized in the last paragraph of Discussion.

Thank you very much